# Simulation Study on Expansive Jet Formation Characteristics of Polymer Liner

**DOI:** 10.3390/ma12050744

**Published:** 2019-03-04

**Authors:** Jianya Yi, Zhijun Wang, Jianping Yin, Zhimin Zhang

**Affiliations:** 1School of Mechatronic Engineering, North University of China, Taiyuan 030051, China; yijianya513@126.com (J.Y.); yjp123@nuc.edu.cn (J.Y.); 2School of Materials Science and Engineering, North University of China, Taiyuan 030051, China; zhangzhimin@nuc.edu.cn

**Keywords:** polymer material, shaped charge liner, expansive jet, tandem warhead

## Abstract

An ideally shaped charge warhead is an effective weapon against armored targets. The use of the gathering energy effect generated by the explosion drives the liner to form a shaped charge jet, which can penetrate the armored target. Existing shaped charge warheads often use a metal liner. Herein, we discuss the characteristics of polymer liners. To study the characteristics of the expansive jet formed by the polymer liner, three polymer materials with different properties—polytetrafluoroethylene (PTFE), nylon (PA), and polycarbonate (PC)—were selected. Using the smooth particle hydrodynamics (SPH) method and the AUTODYN finite element software, the expansive jet formation by the polymer liners was simulated and verified by experimental data. The polymer jets of the different materials exhibit a certain degree of expansivity; however, due to differences in the material properties, the expansive diameter of the jet and the degree of head compaction differed. The expansive diameter of the PA jet was the smallest, and that of the PTFE jet was larger than that of the PA jet, but with a smaller compactness. The PC jet exhibited the largest expansive diameter and the highest degree of compactness.

## 1. Introduction

The technology underpinning armor and anti-armor devices has driven their mutual development. Currently, shaped charges are effective for damaging advanced protective armor and are used in artillery, rocket, and missile warheads. However, with ongoing progress in the development of novel materials, technologies, structural designs, and protective devices, target defensive performance and survivability have been continuously improved. For example, modern heavy armored vehicles are equipped with multi-layer explosive reaction armor to effectively destroy the shaped charge jet or limit the jet-induced damage to the main armor. Previous research studies about shaped charge jets often focused on metal materials, since metal jets can destroy most of the armored targets, but for explosive reaction armored targets, the damage ability of metal jets is greatly reduced. Based on these requirements, novel liner materials have been developed, including glass, ceramics, and polymers; among these, polymer liners have been the most extensively investigated. Metal liner studies have been performed for dozens of years, and their jet characteristics and formation process are well understood. However, metal jet formation theory cannot sufficiently explain the characteristics of polymer jets, because the state changes of the polymer material during explosion are unclear.

As a key factor of a shaped charge jet, the research into liner materials can be divided into four categories of materials: metal, composite, glass, and polymer. As the most classical liner material, metal has been used in the preliminary stage of the discovery of the Mohaupt effect [1]. Nowadays, the performance of the metal jet has been difficult to improve; therefore, improving the purity of metal has become a new topic. The metal liner materials, including copper, molybdenum, tungsten, iron, and titanium, have been researched widely. Baker et al. [2] first discussed the application of molybdenum material in shaped charge, and the molybdenum jet has a very high speed and good toughness with a continuous jet length of 1 m. 

Composite liners are often fabricated by metal composite materials and non-metal composite materials [3]. Metal composite liners are made of several kinds of metal powders, and the typical composite materials are tungsten and copper, owing to the high density of tungsten and the toughness of copper. Bransky et al. [4] reported a metal composite liner by mixing the tungsten–copper powder and tungsten–nickel–iron powder, which improved the penetration performance of the composite liner due to the excellent ductility of the tungsten–copper jet. The non-metallic composite liners are composed of metal and non-metallic materials, which play the role of matrix, and the typical active liner materials are obtained by adding aluminum powder in polytetrafluoroethylene (PTFE) powder. 

With the development of inorganic materials, glass has attracted considerable attention. In 2004, Cowan et al. [5] illustrated the application of oxidized glass in liner materials. The main research results demonstrated that there is no necking phenomenon of the metal jet during the stretching process of the oxidized glass jet. Walters et al. [6] reported a metal–glass composite liner in 2007, indicating that the jet formed by this material is similar to a metal powder jet using the pulsed X-ray technique. However, the penetration experiment results showed that the formation of this kind of jet is asymmetric, and the size and distribution of jet particles are non-uniform. In 2011, Baker et al. [7] studied the jet formation process of glass liners with different densities. The toughness and radial dispersion behavior of glass jets in different regions are observed by pulsed X-ray technique. It is considered that the ductility of the jet is likely related to pressure and glass transition temperature; besides, the particulate behavior of the jet is likely relevant to the brittleness of glass materials. In 2018, Ding et al. [8] adopted a low-density material as the liner of a shaped charge pre-warhead to destroy explosive reactive armor, and the effectiveness was proven by numerical simulation. Therefore, it is possible to achieve the perforation of the explosive reactive armor using the low-density shaped jet, and the low-density materials included float glass, Lucite, and Plexiglas. 

The application of polymer materials in shaped charge was first mentioned by Haney et al. [9] in 2005. They proposed a double liner by adding a layer of polymer lining between the metal liner and the explosive in the common shaped charge, which is equivalent to a double liner. Meanwhile, during the metal jet formation process, the polymer lining would flow into the cavity, where it will burn and decompose to produce gaseous and non-gaseous substances. Especially, if aluminum, titanium, or magnesium is chosen for the material of metal liners, the interaction reaction of metal and polymer will be powerful, and they will become energetic damage elements, resulting in a significant improvement of the damage performance. In terms of damage effect, Baker [10], Daniels [11], and Xiao et al. [12], through the test of a PTFE-based energetic liner shaped charge damage concrete target, found that the jet will have an implosion effect while penetrating the target, and enhance the damage power. Zhang et al. [13,14] carried out a simulation test of PTFE-based energetic jet damage to a steel target, and studied the aftereffect overpressure characteristics of an energetic jet. In the aspect of jet forming, Wang et al. [15] simulated the jet-forming process of a PTFE/Al liner by the Euler algorithm, and carried out an X-ray photography experiment. It was found that the diameter of the energetic jet was thicker, and the ductility was worse, than that of the metal jet. In 2008, Hirsch [16] et al. proposed a liner using the thermoplastic polymer material with high compressibility. The damage performance of a polymer jet was studied experimentally. This kind of jet can expand at the bottom of the hole on the target plate and produce an explosive effect when the target is a concrete target. Nevertheless, the research results only showed that this special effect is related to the setting of stand-off, and there is no further study on the selection of polymer materials. Helte et al. [17] started to research the selection of polymer materials in 2011, and they focused on glass, powder aluminum, copper, and alumina ceramics. The jet morphology formed by different materials adopting the same shaped charge structure was observed, and the expansion characteristics of different degrees were found. Besides, the impact explosion reaction armor experiment was carried out. The results show that glass, powdered aluminum, and alumina ceramics jets can penetrate the explosive reactive armor without explosion. Moreover, the alumina ceramic jet would generate the largest holes on the target plate among the three kinds of materials; thus, it can be used in the shaped charge pre-warhead. Chang et al. [18,19,20] researched a polymer jet penetrating explosive reactive armor, showing the efficient destruction of explosive reactive armor. The diameter of hole was increased by approximately 30% and 70% for the back plate and panels, respectively.

Based on the aforementioned research studies about polymer materials, as shaped charge liner materials, polymer materials have great potential and a promising future in the shaped charge field. In this paper, the characteristics of jet formation in a polymer liner are studied, which could provide a theoretical basis for the application of polymer materials in shaped charge warheads.

## 2. Numerical Simulation

### 2.1. Smoothed Particles Hydrodynamics

Using the AUTODYN-3D software (Ansys 16.0, Canonsburg, PA, USA), polymer jet formation was numerically simulated using the smooth particle hydrodynamics (SPH) method [21]. SPH is a meshless Lagrangian hydrodynamic method that uses particles to solve large deformation and movement discontinuity problems, including high velocity impact welding and metal jets formed by shaped charges [22]. In the SPH method, a set of particles is used to represent an object [23]. The governing equations of the derivatives of the conservation laws were discretized using the following integral:(1)$$f\left(x\right)=\int_{-2h}^{2h}f\left(x^{\prime }\right)W\left(x-x^{\prime },h\right)dx^{\prime }$$
where vectors *x* and $x^{\prime }$ describe the location of the particle of interest and its neighboring particles within the smooth length *h*. In addition, *h* determines the number of particles that affect the interpolation of a particular point. In this method, nearby particles are more influential compared with those further away. The derivative is integrated using the kernel function *W*, which is similar to the shape function in the finite element method, without the need for connectivity between particles. Therefore, the particles can move in the simulated area, allowing a numerical prediction of the jet and detonation products during the explosion.

### 2.2. Finite Element Model

The geometric and numerical models of the shaped charge are shown in Figure 1. The diameter of the charge was 37 mm, the length-to-diameter ratio of the charge was 1, and the liner thickness was 3 mm, with a liner angle of the cone of 60° in the central point initiation mode. The SPH method was used to simulate the jet formation process and continuous stretching and lengthening of the explosive impact of the polymer liner, as shown in Figure 1. The simulation model mainly included explosive and polymer particles. First, a three-dimensional Lagrangian simulation model is established; then, the Lagrangian mesh is deleted, and SPH particles of different substances are filled to complete the calculation. The particle size of the explosive material was set to 2 μm, and 64,389 particles were used. The polymer particle size was set to 3 μm, and 10,266 particles were used. This simulation was performed to study the characteristics of the shaped jet formed by the explosion of polymer materials.

### 2.3. Modeling Material

#### 2.3.1. Liner Material Model

In this study, the Shock state equation was selected to describe the polymer material, which is based on the Gruneisen equation of state of the Hugoniot curve:(2)$$P=\frac{\rho_{0}c_{0}^{2}\eta }{\left(1-s\eta \right)}\left(1-\frac{\gamma_{0}\eta }{2}\right)+\gamma_{0}\rho_{0}E_{m}$$
where $c_{0}$ and s are the parameters of the material shock adiabatic line; $\eta =1-\rho_{0}/\rho$ is the positive volume strain; $E_{m}$ is the initial internal energy, which is usually set to 0 in numerical simulation; and $\gamma_{0}$ is the Gruneisen coefficient. Table 1 shows the shock equation of the state parameters for the three polymer materials tested herein.

In the numerical simulation of the shaped charge jets, the Johnson–Cook and Von Mises strength models are often used, as they can accurately describe the formation of metal jets. However, for polymer materials, the effects of strain hardening, strain rate sensitivity, and melting should be fully considered. Unfortunately, the Von Mises strength model does not consider the hardening effect, and the commonly used Johnson–Cook strength model is limited to a description of metal materials. Therefore, the Johnson–Cook strength model parameters were fitted using the experimental data of material mechanical properties, which were then used for the numerical simulation of polymer expansive jet formation. Table 2 lists the strength model parameters required for the polymer material simulation.

#### 2.3.2. Main Charge Material Model

The Jones–Wilkins–Lee (JWL) equation of state (EOS_JWL) was used to describe the material properties of the CompB explosive. The EOS_JWL accurately describes the volume, pressure, and energy characteristics of gas products during detonation, and is expressed as follows:(3)$$P=A\left(1-\frac{\omega }{R_{1}V}\right)e^{-R_{1}V}+B\left(1-\frac{\omega }{R_{2}V}\right)e^{-R_{2}V}+\frac{\omega E_{0}}{V}$$
where *A*, *B*, *R*_1_, *R*_2_, and *w* are material constants; *V* is the initial relative volume; *E_0_* is the initial specific internal energy; and the C-J parameters include initial explosive density *ρ**_0_*, detonation pressure *P_CJ_* and detonation speed *D*. The specific parameters used in this study are listed in Table 3.

## 3. Results and Discussion

### 3.1. Comparing the Difference between Polymer and Metal Jet

PTFE (polymer) and copper (metal) were chosen, and their jet formation processes were simulated using the smooth particle hydrodynamics (SPH) algorithm. Figure 2 shows the formation process of the polymer and metal jets, where it is clear that the PTFE material can form a continuous jet with good toughness, but the jet morphology differed significantly from that of a typical copper jet. The most significant difference was the jet diameter, which likely originated from the excessively large radial velocity of the PTFE jet that resulted in the radial expansive of the jet during stretching.

The formation process of the two jets was analyzed, and the detonation wave formed by the explosion of the PTFE liner and its detonation products was crushed to form the jet when the charge exploded at 10 μs. The copper liner underwent the same changes, but due to the different properties of the liner materials, differences in jet morphology appeared. The diameter of the PTFE jet was significantly larger than that of the copper jet, indicating that the two liners were in the same state during the crushing process, and can be crushed and collided axially at the set speed. However, after passing through the high-pressure disturbance zone, the PTFE jet exhibited a larger radial expansive velocity, which increased the jet diameter. At 20 μs, the copper jet stretched and lengthened continuously, but the head became increasingly thin, while the PTFE jet expanded continuously in the radial direction. Since the polymer is a tough material, the PTFE jet stretched and lengthened continuously. At 30 μs, the diameter of the PTFE jet head stabilized, while that of the copper jet head broke into multiple particles with the rest of the jet still moving.

From the density distribution chart shown in Figure 3, it is clear that the unbroken part of the copper jet remained a condensate jet, and the density of the jet was largely unchanged. Only the density of the jet part that was broken into multiple particles decreased, but the density of the PTFE jet was very high only near the axis, and its radial distribution density continually decreased. This indicates that the PTFE jet does not form a typical condensation jet under the influence of radial velocity, but instead forms an expansive jet, which is a jet composed of uniformly distributed particles. With increasing time, the head of the copper jet breaks continuously to form smaller particles, while the diameter of the PTFE jet remains largely unchanged. From Figure 3, it is clear that no difference in the density distribution of the slug could be observed between the PTFE and copper jets, suggesting that under the same charge structure, the liner material only affects the shape of the jet without influencing the slug shape.

Due to the intense incandescence produced by the explosion of the shaped charge explosive, visible high-speed photography cannot penetrate the explosive to shoot the jet. To study the behavior of the polymer expansive jet, its formation process was observed by pulse X-ray technology, and the morphology was subsequently obtained. In the experiment, two 450-kV pulse X-ray machines (HP Co., Palo Alto, CA, USA) were used for combined shooting. The experimental principle is described further in Figure 4. The two pulsed X-ray tubes were arranged at a certain angle, and the shaped warhead was arranged vertically on the intersectional axis of the two X-ray tubes to ensure that the shaped jet flowed through it. By controlling the output time of the pulsed X-ray machines, two photographs of the jet morphology at different times were obtained on the X-ray photographic negative [24]. The liner in this paper is manufactured by mechanical processing of bars.

From the X-ray test results of Figure 5, it can be seen that after detonating the shaped charge, the PTFE liner collapsed under the detonation wave and closed to the central axis, subsequently forming a PTFE jet by symmetrical plane collision. Although the shape of the PTFE jet changed with time and showed strong cohesion before 20 μs, with increasing time, the PTFE jet head material gradually showed the characteristics of radially dispersed particles. At a longer time point, it became increasingly obvious that the PTFE jet head material particles followed a radial dispersion. The optical density of the PTFE jet material particles simultaneously increased along the radial direction. The morphology of the PTFE jet was no longer a condensate jet after 40 μs, when the collapse ring formed, indicating that the PTFE liner was crushed and in the powder state. After 70 μs, the PTFE jet particle characteristics were more obvious, and the PTFE slug became significantly large. However, the PTFE jet motion after 100 μs exhibited strong uncertainty characteristics in the helix and offset parameters.

Figure 6 shows a comparison between the results of the X-ray pulse experiment and numerical simulation. The structure of the shaped charge that was used in the experiment was the same as that used in the numerical simulation. The diameter of the head of the PTFE jet was very large, showing a radial expansive state without breaking into a particle jet. In contrast, the copper jet diameter was very small and gradually broke into a granular jet during the stretching process, which is consistent with the numerical simulation results obtained using the SPH algorithm in terms of jet morphology.

### 3.2. Analysis of Jet Formation Performance of Different Polymer Liner

To date, few applications of polymer materials for the gathering energy effect have been demonstrated, and the response of polymer liners during explosions remains unclear. Therefore, three polymer materials with different characteristics were selected as liners to further study the expansive characteristics of polymer jet formation and the influence of polymer material properties on expansive jet characteristics. The feasibility of the numerical simulation method was verified in the previous section. Therefore, the numerical simulation models of the three polymer materials were established, as shown in Figure 7.

Since the density, sound speed, and compressibility of the three polymer materials differ, their specific parameters are shown in Table 4. The key polymer material characteristics that influence expansive jet performance can be studied through the simulation of polymer jet formation and the determination of jet performance parameters. Figure 8 shows a jet-forming process diagram of the PTFE, PA, and PC liners generated by the SPH algorithm.

From Figure 8, it is clear that the jets formed by the different polymer liners exhibited a certain degree of expansibility, but due to the different material properties, the expansive diameter and head density of the jets differed. The performance parameters of different polymer jets are shown in Table 5. The expansive diameter of the jets increased in the following order: nylon (PA) ˂ PTFE ˂ polycarbonate (PC), with the density following a similar trend. According to the rule of metal jet formation, density and sound speed are the key factors that affect jet velocity. When the charge structure is identical, the jet velocity always decreases with increasing liner material density. High sound speed is also necessary for the formation of a high-velocity jet.

From Figure 9, it is clear that the trends for polymer jet velocity are the same as those observed for metal jets, showing an initial increase and subsequent gradual decrease before reaching a stable value. Due to the low density of the polymer material, the velocity of three polymer jets is higher than that of copper jets under the same charge. The velocity of the shaped jet $V_{PA}>V_{PC}>V_{PTFE}$ was determined by the density of the liner material $\rho_{PA}<\rho_{PC}<\rho_{PTFE}$, which was similar to that of metal jet formation.

The expansive characteristics of the polymer jets were mainly reflected in the head of the jet. The expansive ratio can be used to define the degree of jet expansion, and it is the ratio of the diameter of the expanded jet to the diameter of the stable unbroken jet. Since metal jets do not expand, the expansive ratio of the copper jet is one.
(4)$$\gamma =R_{D}/R_{C}$$
where γ represents the degree of expansion, $R_{D}$ is the average diameter of the expanded jet portion, and $R_{C}$ is the average diameter of the continuous jet. The simulation results of the polymer jets yielded expansive ratios of 2.68, 4.6, and 4.7 for the PA, PTFE, and PC jets, respectively, confirming that the expansive ratio of the PC jet is the largest. Figure 10 shows the diameters of three polymer expanding jets at different times.

From Figure 10, it is clear that in the initial stage of jet formation, the expansive diameters of the three polymer jets were similar, but all were much larger than that of the copper jet. This indicates that the part of the liner material flowing out of the high-pressure impact zone determines the jet characteristics. As the jet moves along the axis, the maximum diameter of the copper jet remained unchanged, and the maximum diameter was the same as the initial diameter. However, the three polymer jets lengthened in the axial direction and expanded in the radial direction. The diameter of the PA jet increased slowly, reaching a maximum at 25 μs. The other two polymer jets reached their maximum diameter at approximately 40 μs.

On the other hand, radial expansion of the jet is caused by the radial velocity. The polymer jet exhibited the same radial velocity at the initial stage of formation, which was larger than the metal jet radial velocity. However, due to the different properties of the liner materials, the attenuation degree of the radial velocity varied. The radial velocity of the copper jet decreased to zero at 10 μs, while the radial velocity of the polymer jet decreased relatively slowly, eventually leading to the formation of an expansive jet.

Figure 11 shows that the length of the jets increased linearly, with the polymer jet lengths increasing more than those of the metal jet. Therefore, it can be concluded that the jet length is negatively correlated with the density of the material.

### 3.3. Effect of Liner Thickness on the Performance of Distend Jets

In the classical theory of metal-shaped charge jet penetration, the influence of the jet on target damage is affected by jet density, length, velocity, stand-off, and target strength, among which the most important parameters are the density, diameter, and effective length of the jet. For the metal jet, when the charge structure is fixed, changing the structural parameters of the liner will affect jet performance. For example, when the thickness of the liner is increased, the length and velocity of the jet will decrease, while its diameter will increase. However, for polymer expansive jets, because of their different material properties and formation mechanisms, the existing law of metal jet formation no longer applies. Therefore, the characteristics of polymer expansive jets were studied by varying the liner thickness.

According to the theoretical analysis of jet formation, the liner thickness affects the jet performance to a certain extent during liner collapse. PA, PC, PTFE, and copper were used as liner materials, with the copper jet representing a standard jet. The cone angle of the liner was set to 60° and the liner thickness was varied from 1 to 3 mm. Figure 12 shows the jet morphology of the different liner materials with different thicknesses.

From Figure 12, it is clear that the four materials can form jets with different liner thicknesses, but their morphological properties differ. The three polymer jets exhibited different degrees of expansivity at different thicknesses, while the head of the metal jet did not diverge, but broke into granular jets. The slug of the different materials increased with increasing liner thickness, indicating that the performance of the shaped charge jet was unaffected by the liner material; that is, the liner material only affected the head performance of the shaped charge jet.

Figure 13 shows the morphology of PTFE and copper jets at 11 μs. At the initial stage of jet formation, the morphology of the PTFE and metal jets is similar, and the slug part increased with increasing liner thickness. The head diameter of the PTFE jet was larger than that of the copper jet, but decreased with increasing liner thickness.

From Figure 14, it is clear that the length of the PA and PC jets increased with increasing liner thickness, but the length of the PTFE and copper jets decreased. However, the jet velocity and head diameter decreased with increasing liner thickness, and the extent of this decrease depended on the composition. According to the classical theory of jet formation, increasing the liner thickness will lead to an increased jet diameter, but the diameter of the polymer expanding jet decreased with increasing wall thickness due to flow into the high-pressure area during the collapse and the material compressibility. Figure 15 shows the pressure distribution of the PTFE liner with different thicknesses in the high-pressure jet impact zone.

As can be seen in Figure 15a, with increasing PTFE liner thickness, the pressure in the high-pressure area decreased, with pressure values of 13.14 GPa, 12.66 GPa, and 12.36 GPa, respectively, and the initial jet length and diameter decreased with increasing thickness. From Figure 15b, it is clear that the pressure in the high-pressure area of the PA liner with different thicknesses was 10.02 GPa, 8.51 GPa, and 8.23 GPa, respectively, which were smaller than those observed in the high-pressure area formed by the PC liner. However, with the increasing thickness of the PA liner, the pressure in the high-pressure area decreased, as well as the length and diameter of the initial jet. Figure 15c shows that the pressure in the high-pressure area of the PC liner was 9.21 GPa, 7.11 GPa, and 7.07 GPa, respectively, showing intermediate values between those of PA and PTFE. The change in jet performance as a function of liner thickness was similar to that of the PA and PTFE liners. From Figure 15d, it is clear that the pressure values in the high-pressure area of the copper liner were 33.31 GPa, 25.89 GPa, and 22.8 GPa, respectively, which were much larger than those of the polymer liners. However, the diameter and length of the jet formed by the jet were smaller than those of the polymer jets. Figure 16 shows the pressure curves of the different liners with variable thicknesses in the high-pressure jet collision zone.

According to Figure 16, when the liner thickness is constant, the pressure in the high-pressure area increases with increasing liner material density, which was unaffected by the liner material. When the liner material was constant, the pressure value in the high-pressure area decreased with increasing liner thickness. Therefore, with increasing liner thickness, the initial diameter and length of the polymer expanding jet decreased.

## 4. Conclusions

Herein, polymer materials were used as liners for shaped charge warheads. The formation process of the polymer expanding jet was observed by X-ray photography, which was subsequently analyzed and used to validate the numerical simulation. The results presented herein show that it is feasible to use polymer materials as liners for charge warheads. The SPH simulation algorithm was used to simulate the formation of the expanding jet of the polymer liner. The performance of the expanding jet formed by different polymer materials was analyzed, and the influence of wall thickness was determined. However, the polymer in this paper has not been tested, which will be the focus of my research in the next work. The main conclusions are as follows:(1)The SPH algorithm can effectively simulate the characteristics of the polymer expanding jet.(2)Polymer jets of different materials exhibited a certain degree of expansibility; however, due to the difference in material properties, the expansive diameter and head density of the jets differed. The expansive diameter of the jets was as follows: PA ˂ PTFE ˂ PC; the density followed the same trend.(3)The expansive jet performance of the polymer was largely dependent on the pressure in the high-pressure area. When the thickness of the polymer liner was held constant, the pressure in the high-pressure area increased with increasing liner density, and was unaffected by the performance of the liner material. When the liner material was the same, the pressure in the high-pressure area decreased with increasing liner wall thickness. Therefore, with increasing liner wall thickness, the initial diameter and length of the polymer expanding jet decreased.

## Figures and Tables

**Figure 1 materials-12-00744-f001:**
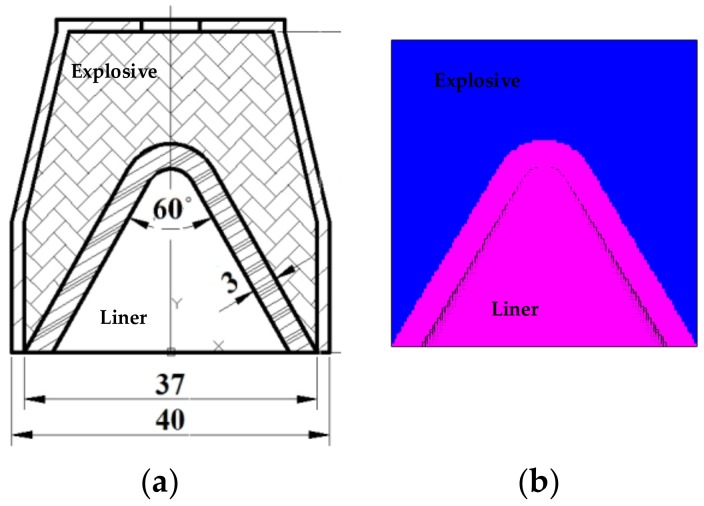
(**a**) Structural sketch of shaped charge warhead; and (**b**) Scheme of simulation model for shaped charge warhead.

**Figure 2 materials-12-00744-f002:**
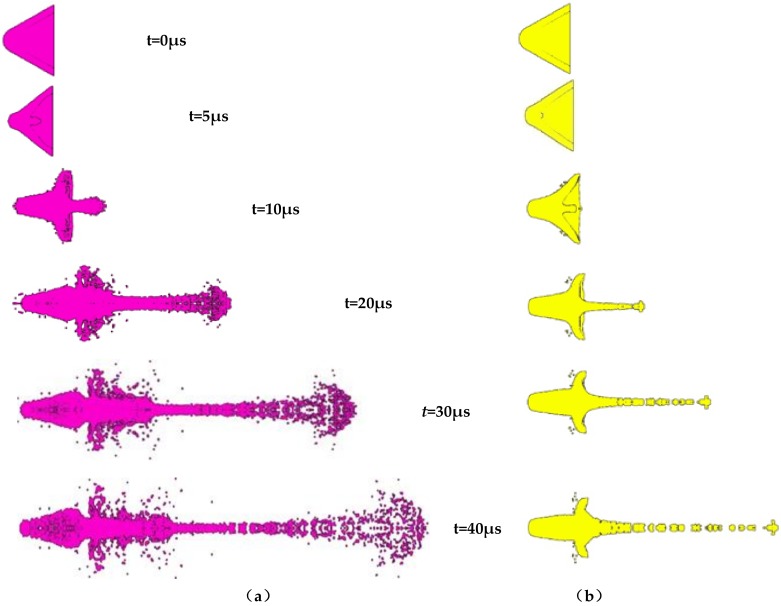
Comparison of the forming process of a polymer jet and a metal jet: (**a**) PTFE; (**b**) copper.

**Figure 3 materials-12-00744-f003:**
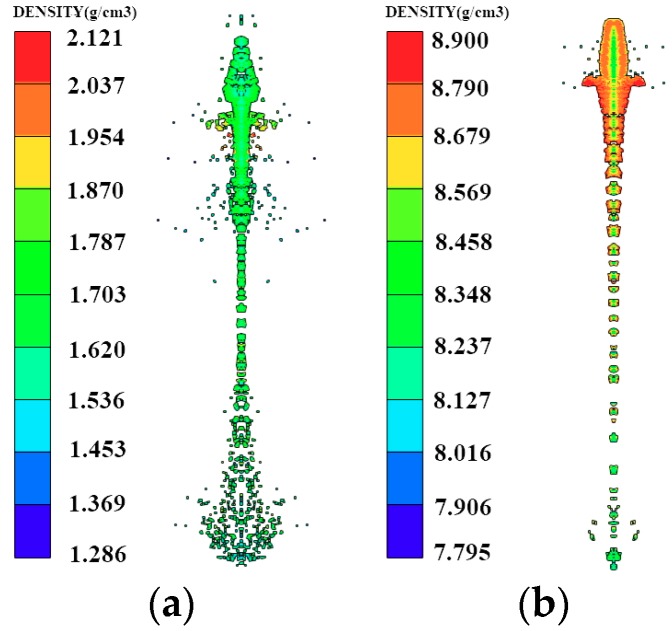
Comparison of the density distribution of polymer jet and metal jet: (**a**) PTFE; (**b**) copper.

**Figure 4 materials-12-00744-f004:**
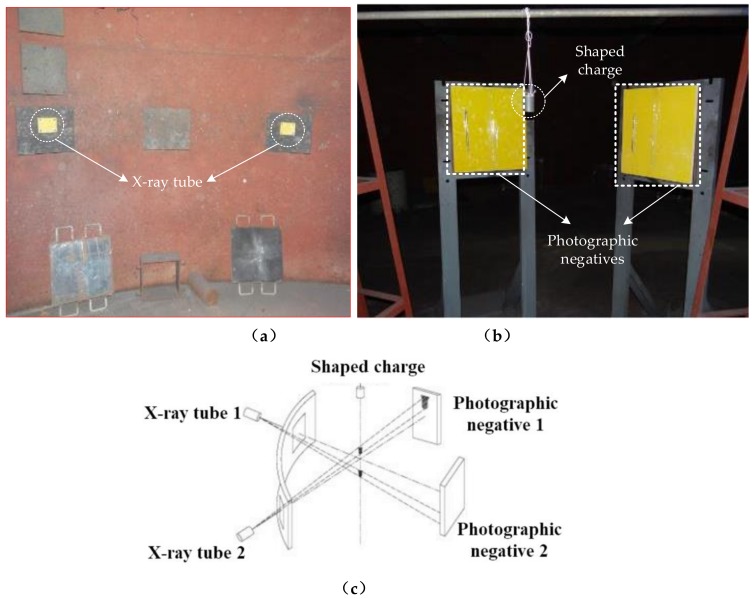
Field layout of pulse X-ray test: (**a**) Test scene graph; (**b**) Test layout; (**c**) Pulse X-ray test principle.

**Figure 5 materials-12-00744-f005:**
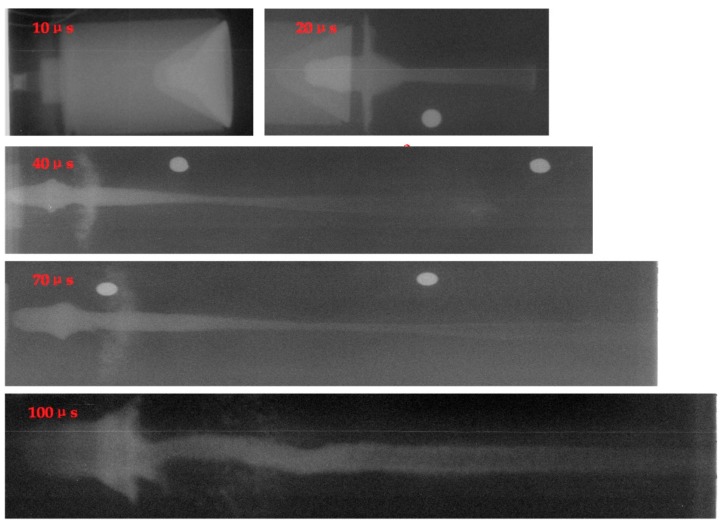
Diagram of test results.

**Figure 6 materials-12-00744-f006:**
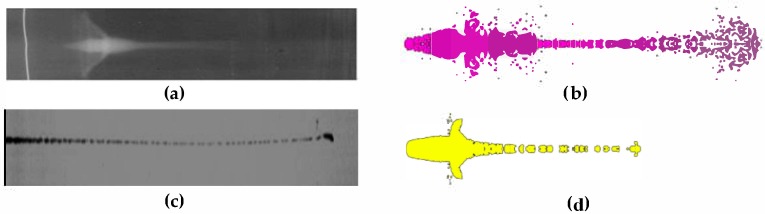
Comparison of numerical results with X-ray pulse experimental images: (**a**) Test results of PTFE jet; (**b**) Simulation results of PTFE jet; (**c**) Test results of copper jet; and (**d**) Simulation results of copper jet.

**Figure 7 materials-12-00744-f007:**
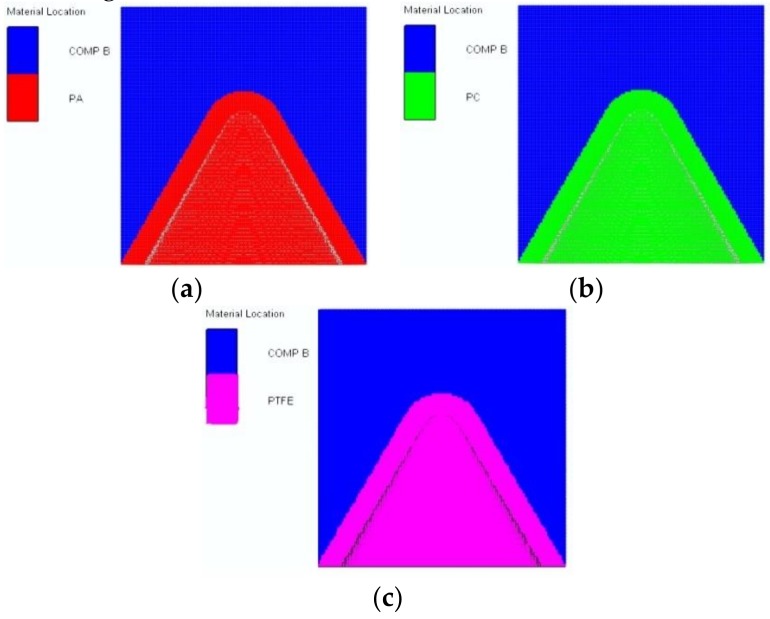
Numerical models for different polymer materials: (**a**) PA; (**b**) PC; and (**c**) PTFE.

**Figure 8 materials-12-00744-f008:**
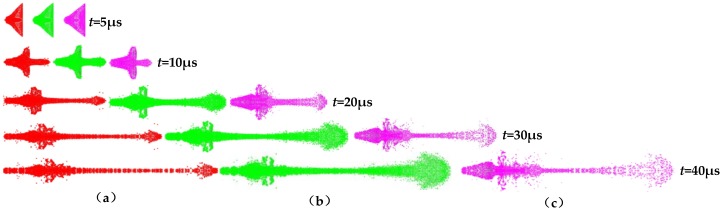
Jet forming process of different polymers: (**a**) PA; (**b**) PC; and (**c**) PTFE.

**Figure 9 materials-12-00744-f009:**
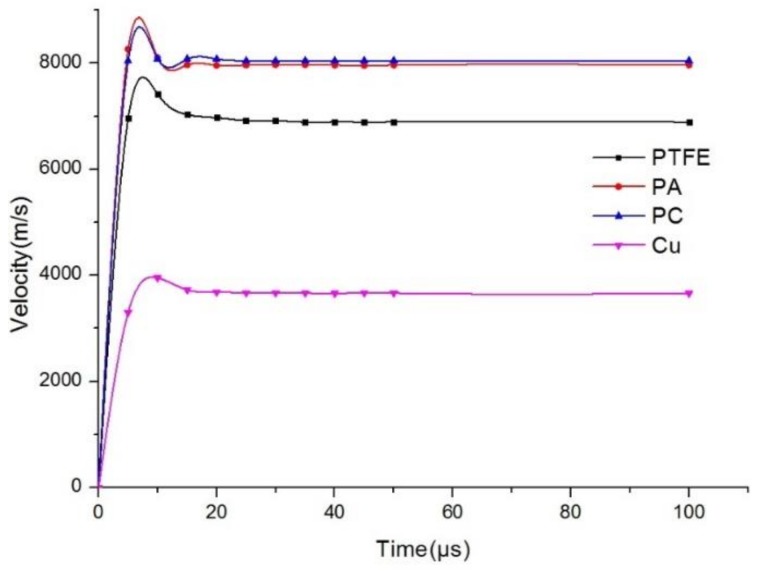
Jet velocity curve of different materials.

**Figure 10 materials-12-00744-f010:**
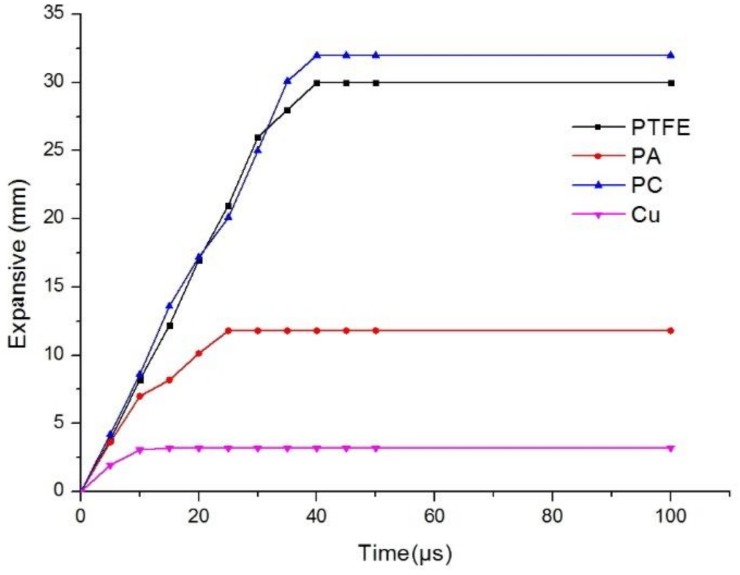
Expansive diameter curve of jets with different materials.

**Figure 11 materials-12-00744-f011:**
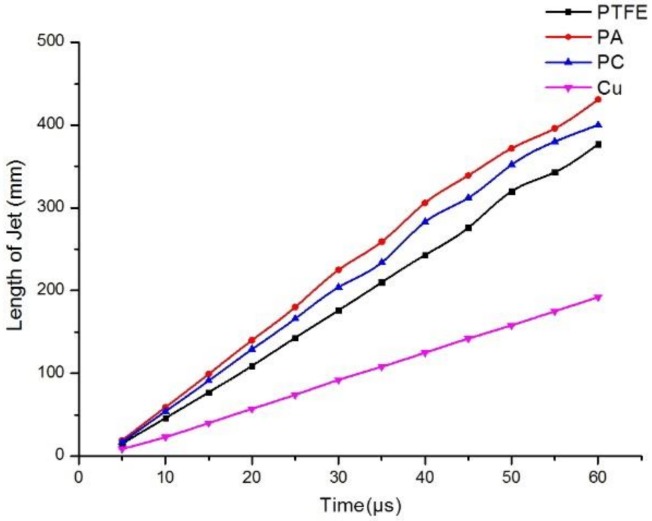
Jet length curves of different materials.

**Figure 12 materials-12-00744-f012:**
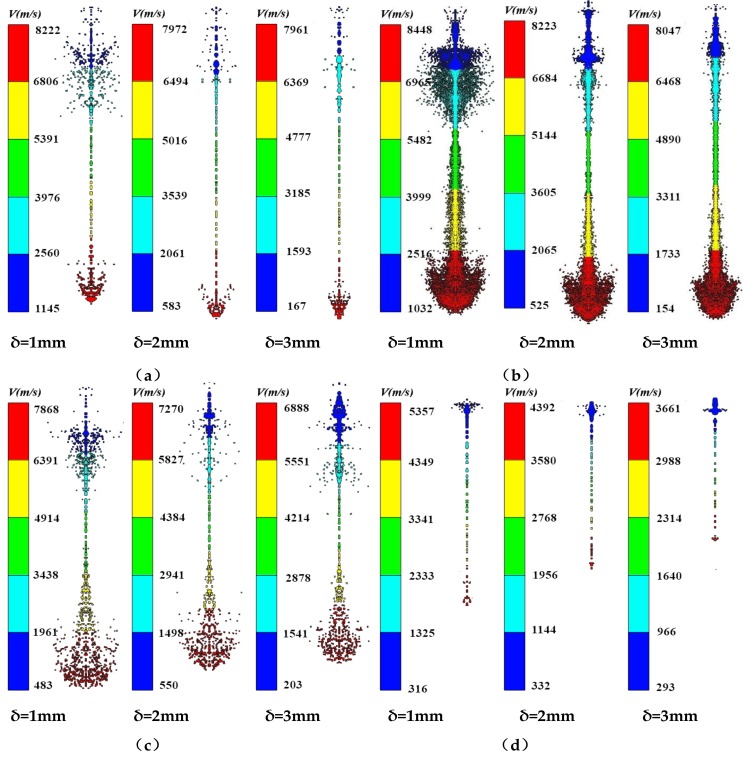
Forming results of different thickness liner jets: (**a**) PA; (**b**) PC; (**c**) PTFE; and (**d**) Copper.

**Figure 13 materials-12-00744-f013:**
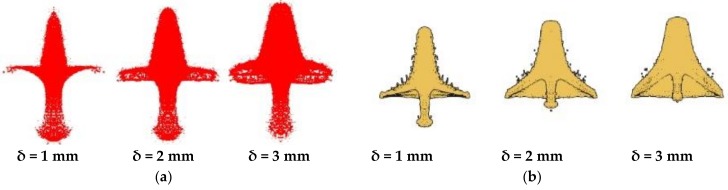
Result diagram of PTFE and copper jets at 11 μs: (**a**) PTFE; (**b**) Copper.

**Figure 14 materials-12-00744-f014:**
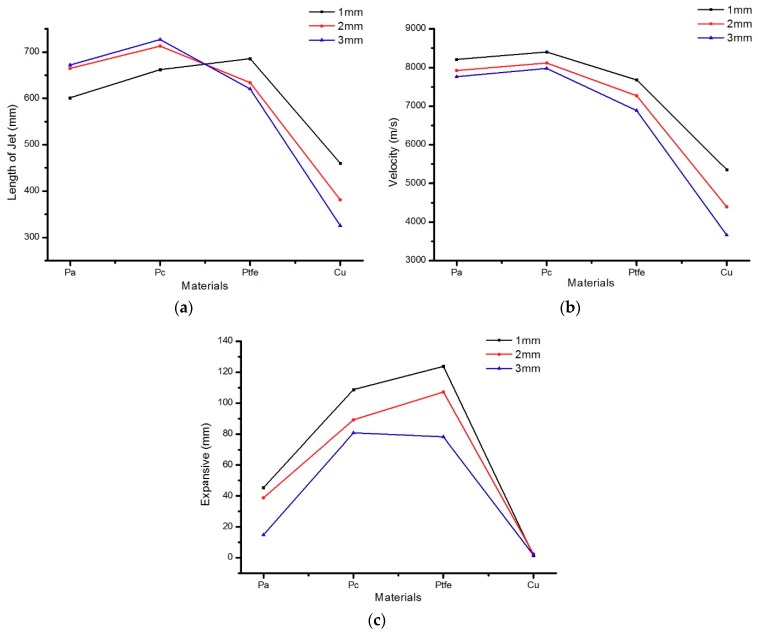
Effect of liner thickness on jet performance: (**a**) The jet length change curve formed by different liners; (**b**) The jet velocity change curve formed by different liners; (**c**) The expansive diameter change curve formed by different liners.

**Figure 15 materials-12-00744-f015:**
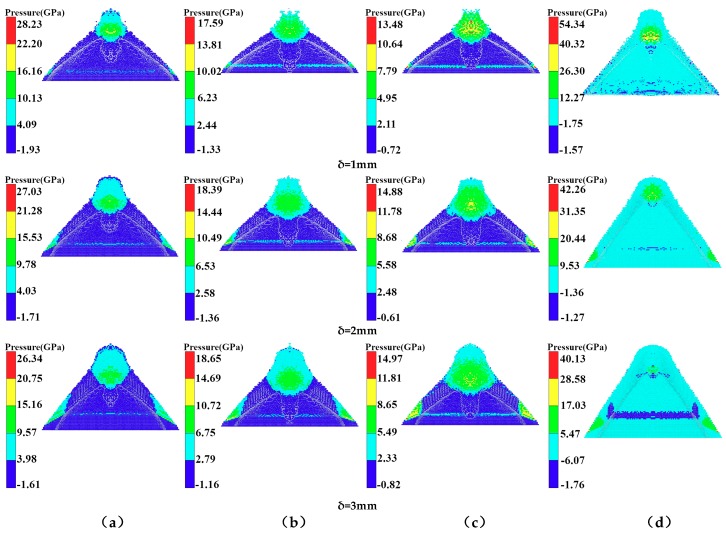
Pressure distribution with different thickness in high-pressure zone of jet impact: (**a**) PTFE; (**b**) PA; (**c**) PC; and (**d**) copper.

**Figure 16 materials-12-00744-f016:**
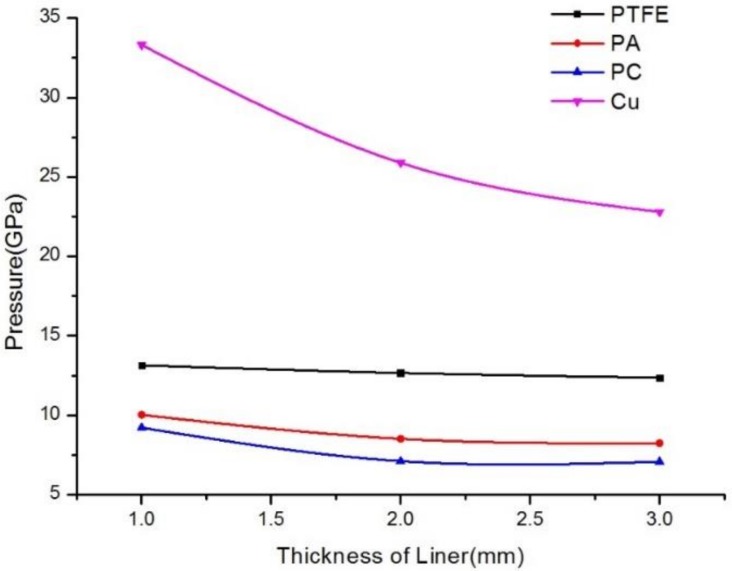
Pressure variation curves of different material liners with different thickness in the impact area.

**Table 1 materials-12-00744-t001:** Parameters of shock equation of state for polymer materials. PA: nylon, PC: polycarbonate, PTFE: polytetrafluoroethylene.

Material	*ρ*/(g/cm^3^)	$\gamma_{0}$	*c*_0_/(cm/μs)	*s*
PA	1.14	0.87	0.229	1.63
PC	1.2	0.61	0.1933	2.65
PTFE	2.16	0.9	0.134	1.93

**Table 2 materials-12-00744-t002:** Strength model parameters of polymer materials.

Material	*G*/(GPa)	*A*/(GPa)	*B*/(GPa)	*C*	*n*
PA	3.68	0.0105	0.0325	0.2570	0.72702
PC	1	0.084	0.03328	0.089	3.1456
PTFE	2.33	0.015	0.038	0.08528	0.3253

**Table 3 materials-12-00744-t003:** The Jones–Wilkins–Lee (JWL) parameters and C-J parameters of the CompB explosive.

Material	*A* (GPa)	*B* (GPa)	*C* (GPa)	*R* _1_	*R* _2_	*ω*	*ρ_0_* (g/cm^3^)	*P_CJ_* (GPa)	*D* (m/s)
CompB	524.2	7.678	1.082	4.20	1.10	0.34	1.717	29.5	7980

**Table 4 materials-12-00744-t004:** Material parameters of different liners.

*Material*	*PA*	*PC*	*PTFE*
***ρ* (g/cm^3^)**	1.14	1.2	2.16
***D* (m/s)**	2290	1933	1340
***G* (GPa)**	3.68	1	2.33
***E* (GPa)**	1.4	2.3	0.28
***Tm* (°C)**	220	267	327
***Tg* (°C)**	40	150	130
***σ_s_* (MPa)**	50	80.6	50
***Compressibility* (GPa^−1^)**	0.0211	0.0906	0.0786

**Table 5 materials-12-00744-t005:** Performance parameters of jets with different materials.

*Material*	*D*/mm	*V*/m/s	*L*/mm
***PA***	14.8	7762	672
***PTFE***	78.2	6884	620.4
***PC***	80.8	7976	727

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
