# Peer review of "Simulation Study on Expansive Jet Formation Characteristics of Polymer Liner"

_materials, 2019, doi:10.3390/ma12050744_

Round 1
Reviewer 1 Report
The autore investigated the characteristics of the expansive jet formed by the polymer liner for three different polymers (PTFE, PA and PC) using numerical simulations (AUTODYN) and experimental testing.
The article is good and results are supported by numerical/experimental data.
Please improve english.
Author Response
Dear Reviewer
Thanks for your letter and the reviewers’ comments concerning our manuscript entitled “Simulation Study on Expansive Jet Formation Characteristics of Polymer Liner” (Manuscript ID: materials-434174). Those comments are all valuable and very helpful for improving our paper, as well as the important guiding significance to our researches. We have studied comments carefully and have made significant changes in the manuscript. The revised version has been uploaded.

Reviewer 2 Report
This paper presents the numerical study of the jet formation from polymer liners utilizing a series of autodyn simulation.
The main approach in this paper is based on the numerical simulation results, and this strongly depends on the accuracy of the simulation results. The authors use a simple comparison of the images from numerical results and testing results with the x-ray, and I believe this is the weakest point of the paper. The comparison of the images is not quite explicit, and may deliver different interpretation depending on how you see it. Most of all, it is not reliable data to further expand to other analysis.
Another issue is that the simulation result from different liner materials are compared only in a few physical parameters (i.e. density, speed, length, etc.) neglecting the importance of the dynamic material properties under the extreme shock condition. The simulation input EoS or the strength model somehow handles this issue, but there is no in depth understanding (or explanation) of the material property during the extreme shock condition, and I don't think this is a proper way to solve the given problem.
After I read this paper, I wasn't sure what I have learned from this, and not sure the scientific contribution in this community.
Author Response

(The authors gave the same response as above.)

Reviewer 3 Report
General comments :
Interesting work in the continuation of previous papers!
There is a general lack of description so that readers are not able at this stage to reproduced the tests or the simulations
From 1st to last page:
- introduction : tens of year instead of hundreds of years;
- background and references in introduction : extend the bibliography for polymer behaviour and numerical modelling especially regarding the phase change, explosive reaction, ...
- Figure 1 : it seems that the model does not represent the shell surrounding the explosive: is it the case and if so explain why and how you can pretend to represent the real case with your model;
- Table 1 and eq 2 : Em is not defined, C0 and s are not given in the table while c1 and s1 appear
- in general : check the text and complete/suppress some sentencies that are uncomplete ""T" page 2, "proposed a double liner by adding", "with high compressive", ...
- in general all over the paper : give scales that will help quantitive estimations and comparisons by the reader that will complete the qualitative comparison between the real tests and simulations, and between simulations: "significantly larger", "streched and lengthedn continuously", Fig 6, ... ;
- in general all over the paper : explain and illustrate where on the deformed geometry, quantities are measured in real tests and in the model (since particles values have no sense in itself) : diameters, pressure, and most of all velocities: for Fig 8 and Fig 9 in particular;
- in models : give the artificial pseudo visocosity values that were used/tuned; explicit wether the simulations are 2D or 3D; explain what kind of integration is used for the SPH derivatives and if there are some corrections in the kernel at boundaries; explain how damage is modeled in particles' behaviour (or influence diameter?) and if phase change or strain rate effects do affect it; explain if you have got some numerical tensile instabilities in SPH models and how you verified if the granular shape of the jet is related to a realistic potential and kinematic energy distribution or not: see for example Fig 2 and 3 and comment "density distribution chart ... unbroken part of the copper ... but the density of PTFE jet ...", comment of Fig 6, Fig 8, ... ;
- Analysis of Fig 14 : not so clear because you use 4 points only: please moderate conclusions;
- Conclusions : please moderate the description of the work concerning the tests because all polymers have not been tested;
Author Response

(The authors gave the same response as above.)

Reviewer 4 Report
Shaped charges with non-metallic liners are known. They could be applied in rare cases. As a rule, penetration of barriers is significantly lower in comparison to shaped charges with metallic liners.
A novelty in the paper are numerical simulations of polymer jet shaping and differences among PA, PC and PTFE jets in comparison to a jet creating from a copper liner. Validation of the forming jets model by experimental data using X-ray flash apparatus enhance the value of the work. Calculated and experimentally registered the phenomena of polymer jet expansion on the path is juicy.
What should be obligatory completed is description of liners manufacture – both polymer and a copper one. Were they pressed, casted, machine cutting or another way ??. The internal structure of a liner usually determines a jet shaping.
Fig. 7 is not necessary, it repeats Fig. 1, which describes all required data.
Author Response

(The authors gave the same response as above.)
